

# Genomics versus mtDNA for resolving stock structure in the silky shark (*Carcharhinus falciformis*)

Derek W. Kraft[1], Emily E. Conklin[1], Evan W. Barba[1],
Melanie Hutchinson[1,2], Robert J. Toonen[1], Zac H. Forsman[1] and
Brian W. Bowen[1]

[1] Hawai'i Institute of Marine Biology, University of Hawai'i, Kaneohe, HI, USA
[2] Joint Institute of Marine and Atmospheric Research, Pacific Islands Fisheries Science Center, NOAA, University of Hawai'i, Honolulu, HI, USA

## ABSTRACT

Conservation genetic approaches for elasmobranchs have focused on regions of the mitochondrial genome or a handful of nuclear microsatellites. High-throughput sequencing offers a powerful alternative for examining population structure using many loci distributed across the nuclear and mitochondrial genomes. These single nucleotide polymorphisms are expected to provide finer scale and more accurate population level data; however, there have been few genomic studies applied to elasmobranch species. The desire to apply next-generation sequencing approaches is often tempered by the costs, which can be offset by pooling specimens prior to sequencing (pool-seq). In this study, we assess the utility of pool-seq by applying this method to the same individual silky sharks, *Carcharhinus falciformis*, previously surveyed with the mtDNA control region in the Atlantic and Indian Oceans. Pool-seq methods were able to recover the entire mitochondrial genome as well as thousands of nuclear markers. This volume of sequence data enabled the detection of population structure between regions of the Atlantic Ocean populations, undetected in the previous study (inter-Atlantic mitochondrial SNPs $F_{ST}$ values comparison ranging from 0.029 to 0.135 and nuclear SNPs from 0.015 to 0.025). Our results reinforce the conclusion that sampling the mitochondrial control region alone may fail to detect fine-scale population structure, and additional sampling across the genome may increase resolution for some species. Additionally, this study shows that the costs of analyzing 4,988 loci using pool-seq methods are equivalent to the standard Sanger-sequenced markers and become less expensive when large numbers of individuals (>300) are analyzed.

Corresponding author
Derek W. Kraft, Kraftd@hawaii.edu

## INTRODUCTION

Many elasmobranchs around the globe have experienced devastating population declines due to overfishing in both target and non-target fisheries (*Musick et al., 2000*; *Clarke et al., 2006*; *Ferretti et al., 2010*; *Heupel et al., 2014*; *Dulvy et al., 2014*; *Oliver et al., 2015*; *Dulvy & Trebilco, 2018*). These species are especially vulnerable to overfishing due to life

history traits such as late maturity, slow growth, low fecundity, and high juvenile mortality, which collectively result in low intrinsic rate of population increase (*Baum et al., 2003*; *Dulvy et al., 2008*). Elasmobranch populations take decades to recover from overfishing, and only if fishing pressure is relieved for an extended period (*Stevens et al., 2000*). Furthermore, many threatened and endangered elasmobranchs have little to no population genetic data that would assist in the resolution of management units (reviewed in *Domingues, Hilsdorf & Gadig (2018)*).

Genetically distinct populations are isolated management units known as stocks; however, stocks can be defined on a smaller scale than genetic populations through other criteria, such as an exclusive economic zone boundry (*Carvalho & Hauser, 1994*; *Ovenden et al., 2015*). Reduced gene flow indicates that if a population is overfished it will not be replenished by immigrants from surrounding populations. This is why managing on a genetic stock-by-stock basis is essential for successful maintenance of exploited species and is sorely needed for over-harvested elasmobranchs (*Dizon et al., 1993*; *Heist, 2004*; *Tallmon et al., 2010*).

For the past two decades the standard for examining population structure in elasmobranchs has been a section of the mitochondrial genome, usually the control region (mtCR) (*Duncan et al., 2006*; *Hoelzel et al., 2006*; *Keeney & Heist, 2006*; *Castro et al., 2007*; *Whitney et al., 2012*; *Clarke et al., 2015*; reviewed in *Domingues, Hilsdorf & Gadig (2018)*). Though recent studies are moving towards multi-marker approaches (*Momigliano et al., 2017*; *Pazmiño et al., 2018*: *Green et al., 2019*), there is still a large body of literature focusing on mtCR. The mitochondrial genome has a higher rate of mutation than most of the nuclear genome (*Brown, George & Wilson, 1979*; *Charlesworth & Wright, 2001*; *Neiman & Taylor, 2009*) and this rate of mutation is a key advantage in vertebrates with slowly-evolving genomes (*Avise et al., 1992*; *Martin, Naylor & Palumbi, 1992*). Elasmobranch mtDNA studies to date have been successful in elucidating population partitions and evolutionary divergences, but the maternal inheritance of mtDNA can limit conclusions about gene flow in cases of sex-biased (usually male) dispersal. Both mtDNA and nuclear markers often have concordant results in sedentary species (*Lavery, Moritz & Fielder, 1996*; *Avise, 2004*; *Zink & Barrowclough, 2008*; *DiBattista et al., 2015*) but, when examined alone, may miss key components of population structure, particularly in migratory fauna (*Pardini et al., 2001*; *Bowen et al., 2005*; *Toews & Brelsford, 2012*). When highly mobile elasmobranchs are examined with both mtDNA and nuclear markers (usually microsatellites), a different picture often emerges in which females are more resident and males are dispersive (*Pardini et al., 2001*; *Schultz et al., 2008*; *Portnoy et al., 2010*; *Karl et al., 2011*; *Daly-Engel et al., 2012*; *Portnoy et al., 2015*: *Bernard et al., 2017*; *Domingues et al., 2018*). Identifying outlier SNPs in the nuclear genome can highlight genes possibly under selection, or show functional responses to environmental changes that have important management consequences (*Jones et al., 2012*; *Fischer et al., 2013*; *Barrio et al., 2016*; *Guo, Li & Merilä, 2016*). Therefore, the combination of mitochondrial and nuclear markers can yield fundamental ecological and evolutionary insights.

High-throughput sequencing is a powerful tool for revealing fine-scale population structure that may be missed by single locus studies (*Andrews et al., 2016*; *Hohenlohe et al., 2018*). However, this method can be costly, especially when examining many individuals as is typical of population genetic or phylogeography studies, and the perceived cost may prevent some from considering a high-throughput sequencing approach. For population genetics approaches based on differences in allele frequencies among populations, equimolar pooling of samples before sequencing is an affordable and accurate strategy for large-scale genetic analysis (*Schlötterer et al., 2014*). Several studies have successfully resolved population structure using a pooled site-associated DNA approach known as pool-seq, including some in commercially valuable marine species (*Gautier et al., 2013*; *Mimee et al., 2015*). Pool-seq provides estimates of allele frequencies for thousands of loci distributed across the genome simultaneously, which in some cases gives greater statistical power that can actually exceed the accuracy of allele frequency estimates based on individual sequencing (*Futschik & Schlötterer, 2010*, but also see *Anderson, Skaug & Barshis, 2014*). Therefore, a comparison of results between the standard mtCR analysis and high-throughput pool-seq is informative in evaluating the relative power and cost of the two approaches for examining population structure.

The silky shark (*Carcharhinus falciformis* (Müller & Henle, 1839)) is the second most commonly harvested shark on Earth (*Rice & Harley, 2013*; *Oliver et al., 2015*). They are one of the top contributors to the shark fin trade and the most common elasmobranch bycatch species in tuna purse-seine fisheries around the world (*Clarke et al., 2006*; *Oliver et al., 2015*; *Cardeñosa et al., 2018*). This pelagic shark, formerly abundant in all tropical oceans, has declined by an estimated 85% in the last 20 years, and is now listed as vulnerable and declining by the International Union for the Conservation of Nature (*Rice & Harley, 2013*; *IUCN, 2017*). Currently silky shark population assessments are conducted at the scale of regional fishery management organization, and conservation management measures are implemented at this scale in the absence of genetic or movement data to define population boundaries. *Clarke et al. (2015)* surveyed silky sharks across these regional management regions and found the western Atlantic was strongly differentiated from the Indian Ocean, but the North Atlantic, Gulf of Mexico, and Brazil could not be differentiated and appeared to comprise a single population. In contrast, using the same mtCR marker, *Domingues et al. (2017)* examined five regions across the Western Atlantic and found the North Western Atlantic was distinct from the South Western Atlantic. The difference between the two studies results from additional sampling in the South West Atlantic from further south than *Clarke et al. (2015)*.

In an era where wildlife management needs far exceed the financial resources to address them, many seek to find the most accessible, robust, and economical means to define management units. In this study, we provide a direct comparison of population genetic analysis methods between Sanger sequencing of the mtCR region and high-throughput sequencing of regional pools of individuals. The same individuals from *Clarke et al. (2015)* were re-sequenced using pool-seq approaches. Regions re-sequenced included Gulf of Mexico, North West Atlantic, and Brazil, as well as one geographically distant location in the Red Sea (Fig. 1). We focused this analysis on SNPs from the mitochondrial DNA as

well as nuclear DNA. We did not analyze any microsatellite loci because they were not a part of *Clarke et al. (2015)*. We then evaluate the economics of conducting pool-seq relative to conventional Sanger sequencing of these same individuals. Ecological and management implications will be addressed in a subsequent companion paper.

## MATERIALS AND METHODS

### Sampling and sequencing

A total of 143 silky shark fin clips or muscle sections were sampled from commercial or artisanal fisheries across four geographic regions and are the same samples examined in *Clarke et al. (2015)*. Specifically, we sampled the Gulf of Mexico (GM, $n = 39$), the North Atlantic (NA, $n = 33$), Brazil (BR, $n = 34$), and the Red Sea (RS, $n = 37$). These sample sizes are slightly lower than *Clarke et al. (2015)*. This reduction was due to DNA degradation over time and the need for high-quality genomic DNA for pool-seq. This is contrary to the DNA quality needed for amplifying a single marker from the mitochondrial control region. Additionally only a subset of the Red Sea samples were randomly selected to keep sample sizes relatively similar.

   DNA was extracted using Qiagen DNeasy Blood & Tissue kit (Qiagen, Mississauga, ON, Canada), following manufacturer protocols. Extracted DNA quality was assessed visually by gel electrophoresis and imaged using Gel Doc E-Z System (BIO RAD, Hercules, CA, USA). Only DNA aliquots with strong genomic DNA bands were further processed, while degraded or overly digested DNA was discarded. Aliquots of high-quality DNA were quantified using an AccuClear Ultra high sensitivity dsDNA quantitation kit (Biotium, Fremont CA, USA) and a SpectroMax M2 (Molecular Devices, Sunnyvale, CA, USA). Libraries were pooled with an equal amount of DNA (ng/µl) contributed per individual to minimize individual contribution bias, totaling 2,000 ng of DNA per library. Numbers of individuals per pool are displayed in Fig. 1. No PCR was performed to ensure individual DNA contribution was kept equal within and across libraries (*Anderson, Skaug & Barshis, 2014*). The rest of the library preparation followed the ezRAD library preparation protocol (*Toonen et al., 2013*; *Knapp et al., 2016*). This included DNA digested with DpnII restriction enzyme and adapters ligated using a KAPA HyperPrep (KAPA Biosystems, Wilmington, MA, USA). Pooled libraries were sequenced using Illumina MiSeq (v3 2x300bp PE) at the Hawai'i Institute of Marine Biology EPSCoR Core sequencing facility.

### Genetic analyses

MultiQC was used to assess sequence quality scores, sequence length distributions, duplication levels, and overrepresented sequences (*Ewels, Lundin & Max, 2016*).
To analyze the mitochondrial genome, a previously published mitochondrial genome from *Carcharhinus falciformis* was used as a reference (GeneBank accession number KF801102). Raw paired-end reads were trimmed with TRIMMOMATIC, mapped to the mitochondrial genome reference BWA (mem algorithm), and variants called using the dDocent bioinformatics pipeline, modified for pool-seq (*Puritz, Hollenbeck & Gold, 2014*,
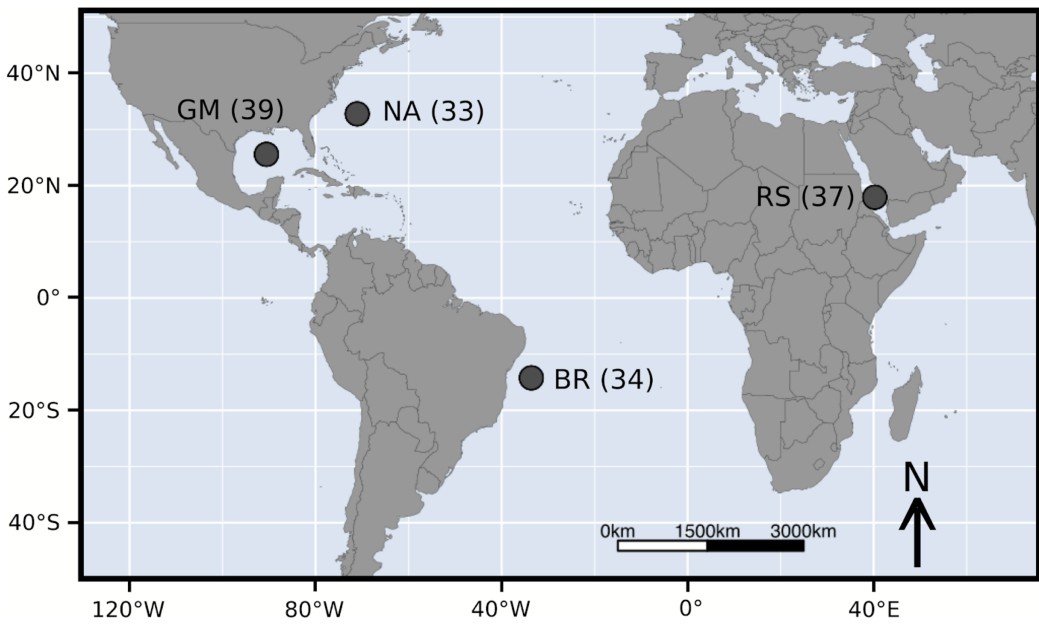

**Figure 1 Sample locations of *Carcharhinus falciformis* followed by sample size.** Abbreviations: GM, Gulf of Mexico; NA, North Atlantic; BR, Brazil; RS, Red Sea.

see below for details). Called SNPs were then analyzed with AssessPool (github.com/ToBoDev/assessPool, see below for details).

The bioinformatics pipeline included dDocent followed by AssessPool. Given that no reference genome was available, a reference was constructed using the dDocent de novo assembly and optimized utilizing the reference optimization steps provided on the dDocent assembly tutorial (http://ddocent.com/assembly/). Before assembly reads were trimmed using default settings and then an overlap (OL) assembly was performed, followed by clustering with CD-HIT with a–c parameter of 90% similarity. For mapping using BWA (mem algorithm) all match, mismatch, and gap open penalty score parameters were also default settings. Different parameters were tested during optimization but did not improve mapping. Within-pool (K1) and between-pool (K2) minimum locus depth values selected for the de novo assembly did impact the results. dDocent provides graphical outputs to help select these values; however, testing a few different values of each is recommended to fully explore the potential of the data by balancing number of contigs by coverage depth (see ddocent.com/UserGuide for details). Selected values for K1 and K2 were 3 and 3 respectively. Once assembled, sequences were mapped, SNPs were called within the dDocent pipeline using FreeBayes, modified for SNP calling in pools (*Garrison & Marth, 2012*, https://github.com/ekg/freebayes). Any contigs that aligned to the mitochondrial genome were removed from this nuclear dataset. The contigs that aligned specifically to the mitochondrial control region were saved for SNP validation to directly compare the results from this pool-seq approach to those previously reported by *Clarke et al. (2015)*.

SNP calling with FreeBayes was optimized for pooled samples using the 'pooled-continuous' option, and minor allele frequency was decreased to 0.05 to capture alleles with frequency greater than 5% in the population (See Supplemental Material for code). The dDocent pipeline outputs SNPs in two variant call format files (.vcf), one with all raw SNPs (TotalRawSNPs.vcf) and another with filtered SNPs (Final.recode.vcf) however dDocent does not optimize filtering for pool-seq data. Therefore, the raw SNPs were processed with the pool-seq specific program AssessPool, which uses VCFtools and vcflib to filter SNPs (Danecek et al., 2011). SNPs were processed with the following filters: minimum pool number of 2, minimum quality score of 20, minimum depth threshold of 30, maximum amount of missing data of 3, maximum allele length of 10, quality score to depth ratio of 0.25 as well as mean depth per site vs. quality score, and finally a maximum mean depth threshold of 1,000 (Table S1). AssessPool then sends filtered SNPs to either PoPoolation2 (Kofler, Pandey & Schlötterer, 2011) or poolfstat (Hivert et al., 2018). PoPoolation2 calculates mean pairwise $F_{st}$ values and significance in the form of $p$-values obtained using Fisher's Exact Test and combined using Fisher's method (as described in Ryman et al. (2006)). Poolfstat (Hivert et al., 2018) takes a different approach, calculating $F_{ST}$ values based on an analysis-of-variance framework (sensu Weir & Cockerham, 1984) to eliminate biases associated with varying pool sizes. AssessPool then organizes, summarizes, and creates visualizations of the data using RStudio (RStudio Team, 2020).

As a quality control test, sequences from Clarke et al. (2015) were downloaded from GenBank (accession numbers KM267565–KM267626), and SNPs from these data were compared directly to SNPs called within the control region of the mitochondrial pool-seq data generated here. Concordance of this validation set of SNPs was determined by Mantel test in R (Legendre & Legendre, 1998) comparing the matrices of pairwise $F_{ST}$ values among populations.

## Cost analysis

The cost of the pool-seq approach compared to Sanger sequencing of individual loci was calculated based on library preparation and sequencing cost at our facility. We did not include labor but calculated the total cost to generate sequence data from each sample included here from such expenses as the extraction, laboratory consumables, PCR amplification, library preparation, reaction clean-ups, quantification, quality control testing, and sequencing costs. These costs were translated into functions in RStudio (RStudio Team, 2020) in which Sanger sequencing is a fixed rate per individual and pool-seq costs are fixed per flow cell on the Illumina MiSeq sequencer MiSeq, but individual cost varies based on number of individuals and number of pooled regions per sequencing run. These functions were then plotted together for comparison.

## RESULTS

A total of 30.8 million reads were generated for the four geographic regions, which averaged 7.7 ± 3.0 million reads per pooled library. Results from the MutliQC assessment showed fairly homogenous output between libraries in regard to sequence quality scores,

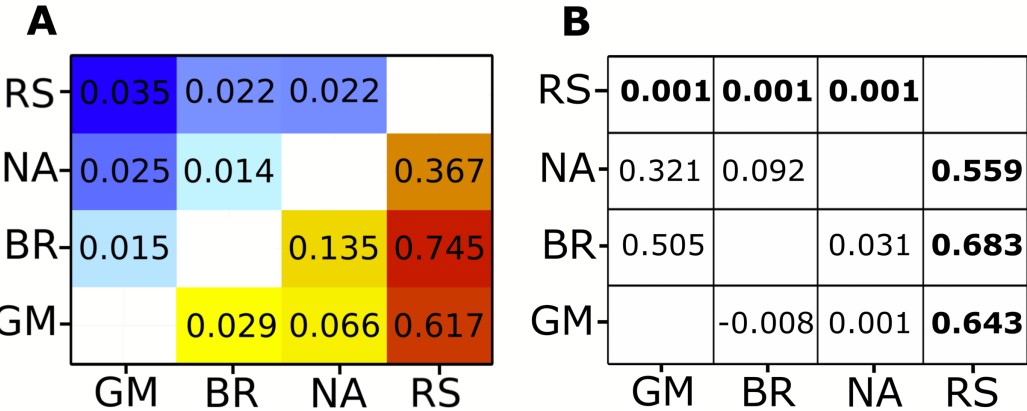

**Figure 2** $F_{st}$ **heat map for mitochondrial and nuclear data compaired to Clarke et al.** $\phi_{ST}$ **and** **P-values.** (A) Pairwise $F_{st}$ values generated by pool-seq methods. Cool colors (top left) are $F_{st}$ values calculated from nuclear genome loci, warm colors (bottom right) are $F_{st}$ values from loci across the entire mitochondrial genome. All pairwise differences are significant ($P < 0.001$). (B) $\phi_{ST}$ results from *Clarke et al. (2015)* on the lower right triangle and *P*-values on the upper right triangle. Significant *P*-values and corresponding $\phi_{ST}$ values in bold. Regional abbreviation are as follows; GM, Gulf of Mexico; BR, Brazil; NA, North Atlantic; RS, Red sea.

GC and per base sequence content, sequence length distributions, duplication levels, overrepresented sequences, and adapter content. Once assembled, aligned, and mapped, 5,792 SNPs were resolved across the mitochondrial and nuclear genomes combined. There were 4,103 biallelic SNPs, 168 multialleleic SNPs and 48 insertions and deletions (INDELs). INDELs and multiallelic SNPs remain a challenge for quantification software, so we restricted our analysis to biallelic loci (*Fracassetti, Griffin & Willi, 2015*). AssessPool creates visualizations of $F_{ST}$ values and allows for visual outlier inspection. No visual outliers were present and given these SNPs are distributed haphazardly across the genome, therefore they are assumed to be putatively neutral.

## Mitochondrial genome

Analysis of the complete mitochondrial genome (17,774 bp) revealed 804 variable sites: 681 biallelic and 17 multiallelic SNPs. Because coverage in this dataset was fairly low on average, most of these SNPs did not meet the filter threshold. After further filtering for the highest quality markers, 30 SNPs were selected to calculate allele frequencies. Pairwise $F_{st}$ values were all significant (Fig. 2; Table S2). The Red Sea had much higher $F_{st}$ values (ranging from 0.367 to 0.745) than any inter-Atlantic comparison (ranging from 0.029 to 0.135). However, all comparisons within the Atlantic still showed significant $F_{ST}$ values, the highest being between the North Atlantic and Brazil, and the lowest between Brazil and Gulf of Mexico (Fig. 2; Table S2).

## Nuclear loci

Our nuclear data showed 4,988 variants, of which 3,422 were biallelic SNPs and 151 were multialleleic SNPs. A total of 346 SNPs remained after the same filtering process for the highest quality SNPs was applied as for the mitochondrial genome. Nuclear markers showed lower $F_{ST}$ values between locations than the mitochondrial data, yet all

comparisons were still significant (Fig. 2; Table S2). The Red Sea showed consistently higher $F_{ST}$ values in comparison to inter-Atlantic comparisons except for the North Atlantic to Gulf of Mexico comparison, which showed the second highest mean $F_{ST}$ value (Fig. 2; Table S2). The highest value ($F_{ST}$ = 0.035) was observed between Gulf of Mexico and the Red Sea, whereas the lowest ($F_{ST}$ = 0.014) was between the North Atlantic and Brazil, which had the highest $F_{ST}$ value within the Atlantic for the mitochondrial data.

### SNP validation

SNPs called in the mitochondrial control region using the pool-seq protocol were compared with those reported in Clarke et al. (2015). Of the 34 SNPs in their study, 14 had a minor allele count (MAC) of less than or equal to 3 and several were singletons. These SNPs are removed from the pool-seq data due to MAC SNP filter of >3 to remove sequencing errors that might be scored as rare alleles during high-throughput sequencing. Therefore, singletons or any rare allele represented fewer than three times in a population will inherently be removed from pool-seq data sets. Fortunately those rare alleles do not tend to overly impact $F_{ST}$ values and should not bias interpretations of population structure (Bird et al., 2011; Toonen et al., 2011). Three SNPs were found in the Clarke study with a MAC of >3 that were not present in the pool-seq data; however, the remining 17 SNPs were all present in our data, plus one that was not found in the Clarke study (Fig. S1). Despite the loss of these rare alleles from the SNP validation set, pairwise $F_{st}$ values estimated by both methods remained highly correlated (Mantel test, $r^2$ = 0.96, $p$ < 0.05), and comparisons between the Red Sea and all three Atlantic populations showed the same relative magnitude between both methods.

### Cost analysis

The findings for cost analysis indicate that pool-seq reaches a threshold at approximately 300 individuals, after which this approach offers cheaper results than individual Sanger sequences. Furthermore, the cost is only twice as expensive at just over 100 individuals (Fig. 3A). The pool-seq approach provides a far higher ratio of information for the cost, yielding greater population resolution. This cost assessment does not include analytical time, labor, or effort associated with pool-seq analyses such as access to computer resources and expertise with bioinformatic pipelines. However, these costs are likely to decrease in the near future as bioinformatic pipelines are improved and become more widely available, for example as applications deployed via cloud based platforms such as Galaxy (https://usegalaxy.org/) or CyVerse (https://cyverse.org/). It is also important to note that the choice of pool-seq methodology has many caveats, which are discussed in greater detail in the 'considerations on pool-seq' section of the discussion below.

### DISCUSSION

Elasmobranchs are being harvested at unsustainable levels in several commercial fishing industries around the world. A fundamental step in successful management of any species is resolving population boundaries so they can be managed on a genetic stock-by-stock basis. As genetic sequencing technologies advance, there is greater opportunity to
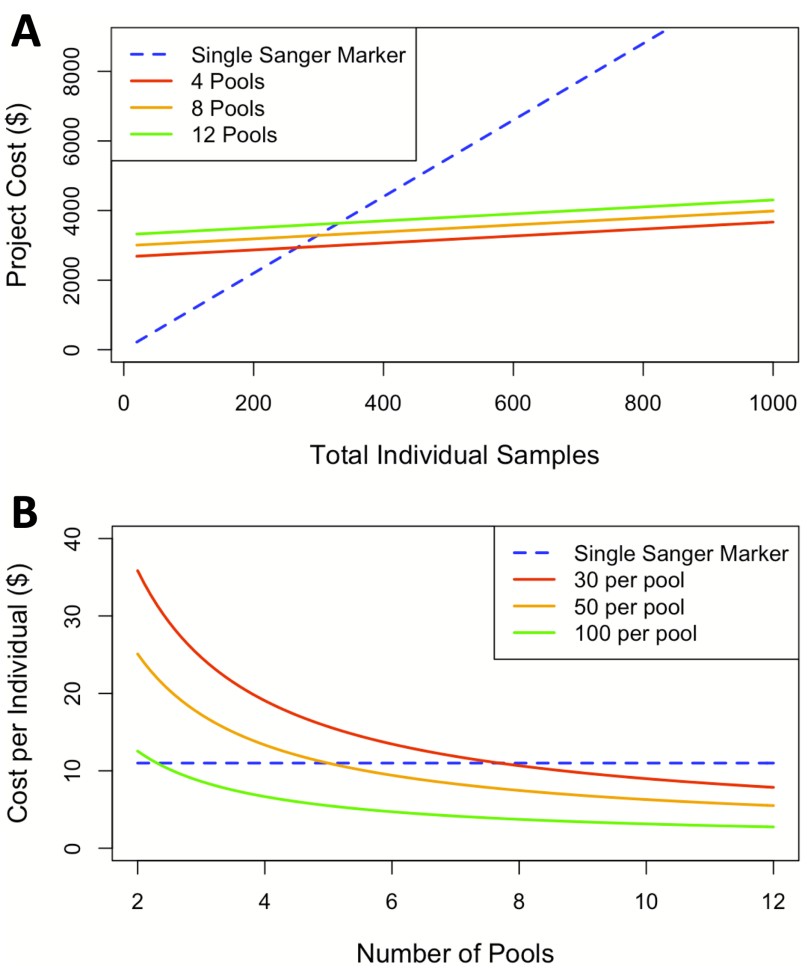

**Figure 3 Cost comparisons between sequencing projects using a single Sanger marker to projects using pool-seq with varying numbers of pools.** (A) Sequencing costs comparing number of individuals to total cost between Sanger at our facility and three pool-seq projects at our facility containing 4, 8, and 12 pools respectively, where pool sizes change with number of individuals. (B) Sequencing cost per individual with fixed pools across different number of Pools.

detect even small-scale genetic differences between populations. When these differences amount to statically significant allele frequencies at the population level, this indicates limited exchange among distinct stocks.

Here, we validate the utility of pool-seq using the same individuals as a previous study (*Clarke et al., 2015*) and show that pool-seq recovers additional population structure relative to Sanger sequencing of the mtDNA control region. Pool-seq was able to detect isolated populations between the Gulf of Mexico, Western Atlantic, and along the Brazilian coast, where *Clarke et al. (2015)* found no population structure. As expected, the Red Sea population was highly isolated from Atlantic conspecifics using both approaches.

One advantage of this pool-seq approach is that we recover SNPs throughout the entire mitochondrial genome along with thousands of additional nuclear loci that together provide greater statistical power to detect finer scale population structure

(*Ryman & Palm, 2006*; *Larsson et al., 2009*; *Kurland et al., 2019*). The pool-seq approach yielded significant genetic structure among inter-Atlantic regions in both mtDNA and nuclear loci, whereas Sanger sequencing of the mtCR lacked power to resolve significant differences among the same populations. The congruence between the mitochondrial genome and nuclear loci reinforces the conclusion of population structure among all regions sampled in this study.

In this case, pool-seq lived up to the promise of increased power to detect fine-scale structure, but does it live up to the promise (*Ferretti, Ramos-Onsins & Pérez-Enciso, 2013*; *Schlötterer et al., 2014*) of being cost-effective? Individual extraction costs remain fixed across both approaches and Sanger sequencing generally has a flat rate per individual, including PCR primers and reagents, and sequencing per individual per locus. In contrast, pool-seq has a flat sequencing cost determined by the number of reads generated from the high-throughput sequencing platform, plus a small additional cost per pool for the exact quantification of DNA for equimolar pooling and the library preparation for high-throughput sequencing. Comparing costs at our institution between a single Sanger sequencing marker and pool-seq on the Illumina MiSeq platform indicates pool-seq becomes less expensive when sample size of the study rises above 300 individuals. Although the cost per pool is essentially fixed, when higher numbers of individuals are included per pool, the price per individual analyzed is further reduced (Fig. 3B). Our comparison here is limited to 12 pools due to the maximum number of reads per lane produced on the MiSeq platform. Therefore, analyzing more than 12 pools would require additional sequencing runs and result in a step increase in the cost per individual/pool, although this would differ among other Illumina machines (such as the HiSeq, NextSeq or NovaSeq) or other high-throughput sequencing platforms (such as the PacBio Sequel II). Larger numbers of pools could be run on some of these machines, but with differing individual read lengths and sequencing depths, which also introduce other trade-offs. Likewise, samples can also be run with individual barcodes, therefore gaining the individual information lost by pooling specimens, but with increased initial setup and sequencing costs. There are so many options by which to apply these methods that we cannot possibly consider them all here, and the availability, cost, and trade-offs associated with each should be ideally considered by individuals when designing high-throughput sequencing projects. In our case, we considered only the options currently available to us through our campus sequencing core, and all these pool-seq price comparisons are to a single Sanger-sequenced marker. Thus, when considering the information acquired from pool-seq compared to the cost from traditional single mitochondrial marker the price per individual advantage is massively amplified.

## Considerations with pool-seq

As with any sequencing technique, there are still several factors to consider before deciding if pool-seq is appropriate for a particular study. Multiple reviews have been published on high-throughput and pool-seq approaches demonstrating pros, cons, and

considerations with these methods, which are beyond the scope of this study. Interested readers should consult *Perez-Enciso & Ferretti (2010)*, *Futschik & Schlötterer (2010)*, *Kofler, Betancourt & Schlötterer (2012)*, *Ferretti, Ramos-Onsins & Pérez-Enciso (2013)*, *Schlötterer et al. (2014)*, *Andrews & Luikart (2014)*, *Andrews et al. (2016)* and *Kurland et al. (2019)*.

Pooling assumes individuals are from the interbreeding individuals within a single population of the same species. Therefore, care needs to be taken to avoid cryptic species, combining multiple populations (Wahlund effect), or other unintentional bias when selecting individuals to pool (*Garnier-Géré & Chikhi, 2013*). For wide ranging pelagic species such as the blue shark or oceanic whitetip it seems reasonable to pool individuals from a larger area than it would be for small benthic species such as horn sharks, wobbegongs, or most rays. Population structure may be obscured if the geographic range per pool is too large or if there is complex population structure (sensu *Bowen et al., 2005*), because individuals from multiple sub-populations will be mixed into a single pool from which allele frequencies are calculated. Certainly pool-seq is not appropriate in all cases. It is a cost-saving approach for analyses based on allele frequencies only, because individual information is lost by pooling, including haplotypes/genotypes and linkage disequilibrium information. Also, pooling makes it difficult to distinguish between low frequency alleles in the population and sequencing error. Therefore, careful filtering must be applied to ensure only valid SNPs are analyzed instead of analyzing sequencing noise (*Anand et al., 2016*; *Schlötterer et al., 2014*). Finally, the estimation of $F_{ST}$ from pooled data remains a subject of some debate, and new approaches and bias corrections are being actively developed (*Kofler, Pandey & Schlötterer, 2011*; *Hivert et al., 2018*). To account for this uncertainty, we include analyses based on both the original PoPoolation2 (*Kofler, Pandey & Schlötterer, 2011*) package and the newer poolfstat (*Hivert et al., 2018*) that explicitly considers potential biases associated with varying pool sizes. The two approaches yield slightly different $F_{st}$ values (see Table S2), however a comparison of the two $F_{st}$ matrices shows strong correlation (Mantel $r = 0.991$ for mitochondrial and $r = 0.978$ for nuclear data, $p < 0.05$). Therefore, only those $F_{st}$ values calculated by PoPoolation2 are reported in the main text for ease of presentation.

Though pool-seq has been shown to be an affordable and reliable tool for population genomics (*Futschik & Schlötterer, 2010*; *Gautier et al., 2013*; *Rellstab et al., 2013*; *Konczal et al., 2014*; *Schlötterer et al., 2014*; *Kurland et al., 2019*), projects with larger budgets could allocate funds for any of a variety of other genomic sequencing techniques such as individual RADseq libraries (*Hohenlohe et al., 2010*), GBS (*Narum et al., 2013*), SNP arrays (*Qi et al., 2017*), bait capture (*Feutry et al., 2020*), or low coverage genomewide sequencing (*Therkildsen & Palumbi, 2017*). These approaches allow for individual genotyping to examine questions that require individual-level information and could provide a deeper assessment of populations. However it is also important to consider not all labs can afford to generate genomic level data, especially in developing countries, and having a cost-effective alternative to single marker studies will continue to be invaluable to many.

## CONCLUSIONS

The finding of population structure on the scale of North Atlantic/Gulf of Mexico/Brazil is nearly unprecedented for a pelagic shark. Population structure in globally distributed sharks is typically detected on a scale of ocean basins (Atlantic versus Indo-Pacific, *Castro et al., 2007*; *Graves & McDowell, 2015*) and a few pelagic fishes have no population structure on a global scale (e.g., Basking shark, *Cetorhinus maximus*, *Hoelzel et al., 2006*; Blue shark *Prionace glauca*, *Veríssimo et al., 2017*; Wahoo, *Acanthocybium solandri*, *Theisen et al., 2008*). The resolution of isolated populations on the scale of North Atlantic Ocean is more typical of coastal species than pelagic species. The silky shark seems to be a pelagic species with a somewhat coastal population structure. This has strong implications for international management because smaller stocks imply smaller populations which are more readily depleted. At a minimum, these data require rethinking a single population management approach for the Atlantic, and this pattern needs to be investigated for this species across the Indo-Pacific as well.

Overall this study demonstrates pool-seq is a powerful and cost-effective tool for analyzing large portions of the genome which the methods traditionally used for elasmobranchs could not supply. Sharks and rays are an imperiled group of species that could benefit from advanced genomic studies to outline appropriate management units. Finally, although the technology is becoming cheaper and easier to apply, it is a common pitfall to assume everyone in the field can afford, or must use, these approaches to produce defensible science. *Bowen et al. (2014)* advocate judicious rather than wholesale application of genomic approaches as the most robust course of study, particularly when considering the global inequities in available research budgets. Sanger sequencing is still more cost effective for small numbers of individuals, but as the number of individuals included in a study rise, the cost per individual reaches the point where high throughput sequencing studies can be cheaper than sequencing a single mitochondrial marker from each individual. We provide an example of just such a case here, and highlight the potential advantage of cost savings together with increased power for resolution of fine scale population structure. Though there is still additional cost of using cluster computer servers and bioinformatics programs, these cost are dropping as technology advances. When study organism and sampling strategies are assessed and implemented into the study design, pool-seq has great promise for augmenting the scientific foundations for management of marine recourses.

## ACKNOWLEDGEMENTS

This study was made possible by the generous donation of specimens by Christopher R. Clarke, Mahmood Shivji, Stephen A. Karl, J.D. Filmalter, and Julia Spaet. We thank members of the ToBo Lab for sharing expertise, advice and discussions that contributed to this manuscript. Special thanks to Darren Lerner, Kim Holland, Carl Meyer, S. Gulak, D. Bethe, D. McCauley, C. Wilson, Guy Harvey Ocean Foundation, and Save Our Seas Foundation. The views expressed herein are those of the authors and do not necessarily

reflect the views of NOAA or any of its subagencies. This is contribution #1821 from the Hawaii Institute of Marine Biology, contribution #JC-15-32 from the Hawaii Sea Grant Program, and contribution #11128 from the School of Ocean and Earth Science and Technology at the University of Hawaii.

### Funding

This paper is funded by a grant from the National Oceanic and Atmospheric Administration, Project R/FM-18, which is sponsored by the University of Hawaii Sea Grant College Program, SOEST, under Institutional Grant No. NA05OAR4171048 (to Brian W. Bowen) from NOAA Office of Sea Grant, Department of Commerce. Additional funding was provided by the National Science Foundation (OCE-15-58852 to Brian W. Bowen). The funders had no role in study design, data collection and analysis, decision to publish, or preparation of the manuscript.

### Grant Disclosures

The following grant information was disclosed by the authors:
National Oceanic and Atmospheric Administration: R/FM-18.
NOAA Office of Sea Grant, Department of Commerce: NA05OAR4171048.
National Science Foundation: OCE-15-58852.

### Competing Interests

Robert J. Toonen is an Academic Editor for PeerJ.

### Author Contributions

- Derek W. Kraft conceived and designed the experiments, performed the experiments, analyzed the data, prepared figures and/or tables, authored or reviewed drafts of the paper, and approved the final draft.
- Emily E. Conklin analyzed the data, prepared figures and/or tables, authored or reviewed drafts of the paper, and approved the final draft.
- Evan W. Barba analyzed the data, prepared figures and/or tables, authored or reviewed drafts of the paper, and approved the final draft.
- Melanie Hutchinson performed the experiments, authored or reviewed drafts of the paper, and approved the final draft.
- Robert J. Toonen conceived and designed the experiments, analyzed the data, authored or reviewed drafts of the paper, and approved the final draft.
- Zac H. Forsman conceived and designed the experiments, analyzed the data, authored or reviewed drafts of the paper, and approved the final draft.
- Brian W. Bowen conceived and designed the experiments, authored or reviewed drafts of the paper, and approved the final draft.

## Animal Ethics

The following information was supplied relating to ethical approvals (i.e., approving body and any reference numbers):

Samples were collected aboard commercial fishing vessels therefore no IACUC approvals were needed.

## Data Availability

Raw data are available at the NCBI Sequence Read SRA: PRJNA647728

## Supplemental Information

Supplemental information for this article can be found online at http://dx.doi.org/10.7717/peerj.10186#supplemental-information.

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
