# Peer review of "Genomics versus mtDNA for resolving stock structure in the silky shark (Carcharhinus falciformis)"

_PeerJ, doi:10.7717/peerj.10186_

## Round 0.1 · original submission · Minor Revisions

· Academic Editor

Minor Revisions

Besides the answers to Reviewers’ comments, I further ask the Authors to comment on the limitations of mtDNA for the purpose of detecting population structure due to uniparental mode of transmission.

·

Basic reporting

The manuscript fully meets PeerJ standards. The english is professional and unambiguous, the figures are well prepared and informative. Raw and supplementary data are included and the previos MER review and the improved version (with track changes) were a nice add.

Experimental design

The research is within the Aims and Scope of the journal. The questions is well defined and relevant. The methods are described in detail and all info is included such as the raw data and the supplementarymaterial.

Validity of the findings

Please check the "General comments for the authors" field.

Additional comments

First of all, congrats for the authors for the manuscript. It is clearly written in professional, unambiguous language. The Figures are informative and well described. I’ve included my considerations/suggestions within the attached PDF file.

In general, this is a really nice and already revised, and improved manuscript. It definitely deserves to be published since it touches a sensible point regarding the recent high-throughput molecular methods, i.e., its cost effectiveness. However, the choice for the pool-seq is not only related with the per sample cost but to several parameters and I suggest to stress this out along the Discussion and also within the manuscript’s Conclusions. However, I finished the manuscript without a clear idea on how much cost a unique sample in both compared techniques and I believe, regarding the scope of the manuscript, that it should be a nice and important result. Ok, it will vary a lot depending the chemistry, platform, laboratory, etc, etc… but I believe we should finish reading this manuscript able to figure out the magnitude of costs per sample we are talking about.

Together with the points presented within the PDF, please check the Competing Interest Statement since there is no CIS, just a sentence saying that one of the authors is a PeerJ Editor. A formal CI Statement should be included prior the acceptance following the PeerJ guidelines for submission. The raw data as well as additional supplementary material are available and are a great source of info. Thanks for providing them.

Finally, I commend the authors for their work and manuscript. Imho this manuscript is good to go.

·

Basic reporting

Kraft et al. present a manuscript entitled “Genomics versus mtDNA for resolving stock structure in the silky shark (Carcharhinus falciformis)”, where they tested the use of pool-seq as an alternative, cost effective and more sensitive tool for the delination of the C. falciformis stock structure. They examined the population structure from four different geographical sites of the Silky Shark C. falciformis. The advantage of this study is that they used the same individuals examined by Clarke et al. 2015, who used sanger sequencing of mtCR of these populations, and therefore were able to compare their findings and limit any introduced variation. Findings showed that the pool-seq revealed similar Fst values to Clarke et al. 2015, but also revealed finer structure which was not picked up by sanger sequencing on mtCR. In my opinion, this manuscript fills an important gap in the field of conservation genetics of elasmobranch, which has been hindered and restricted by the use of few mtDNA and microsatellite markers only. The huge genome size of many elasmobranch species and lack of reference genomes in many cases has also contributed to the limited advancement in this field. I believe this work will contribute towards filling this gap. However, I do have few concerns regarding few points in the manuscript, I listed these concerns based on scale and mentioned them in two sections; major and minor comments as listed section 1.3.

1) Structure and Criteria:

1.1: Basic Reporting:
The manuscript is written in clear English and follows a logical flow. Authors introduced the subject well and transition points were well in place with sufficient referencing to the needed background.

Experimental design

1.2: Experimental design:
The manuscript has clearly stated the research question, the gap intended to fill which fits within the scope of PeerJ. Methods are well explained, reproducible but need to provided more information to evaluate the findings (which I mention in the major and minor comments).

Validity of the findings

1.3: Validity of the findings:
The data are made available in repository. I was happy to see that the data were analyzed and validated using two different systems in measuring Fst, which seems to be a center of debate in pool-seq. These two methods were the poolfstat and popoolation. The conclusions are clearly stated and connected to the main question investigated.

In general, the study is well presented. However, I do think the manuscript still posses some parts that needs clarification and perhaps modifications. In the general comments for the authors section, I will present the major comments followed by another section that is more specific and minor.

Additional comments

- Major comments:

1) Authors have noted in their manuscript that in order to reduce cost, it is important to have a large number of samples in the pool, I think its important to also mention to the readers what challenges (e.g. sequencing errors) come with increasing pool size, how it effects the accuracy of allele frequencies and may be list some reference that mentions solutions to handle this challenge. May be add a few sentences in the discussion to highlighting this point and to handle it.

2) Identification of rare variants: Rare variants are very useful in identifying fine-scale structure, and I think it would add more strength to your argument of using pool-seq with elasmobranchs, given that population structure in most elasmobranchs is determined at a fine scale. In your study you have chosen to set your MAF to 5% (Methods, in line-189), which has very likely removed informative rare variants and possibly some private variants that could be very useful in identifying fine scale population structure. I understand the risk here is that these rare variants could be confounded with sequencing errors specially in pools with large sample size, generating many false positives. But I have two points that I would like to suggest in here:

a. First; would you consider changing your MAF to 1% instead of 5%, since you pools are composed of a mean of 36 individuals (72 autosomes), you optimum lower detection limit for variant alleles in your pools will be: 1/72 (AF= ~ 0.01).
b. Second; if you decide to change your MAF to 1%, it would be useful to verify the accuracy of MAFs in your pool-seq by using CRISP [Comprehensive Read analysis for Identification of Single Nucleotide Polymorphisms (SNPs) from Pooled sequencing] to remove spurious variants due to sequencing errors yet still incorporate real rare variant alleles (As was presented in Anand et al. 2016, which you also reference in your manuscript). In this paper, they also presented a filtering guideline using Kolmogorov-Smirnov (KS) test, to help validate rare variants from large pools. Have you considered using CRISP to generate quality score for each variant, then use that score to apply a quality based filtering using the Kolmogorov-Smirnov (KS) test.
Paper: Anand, S., Mangano, E., Barizzone, N., Bordoni, R., Sorosina, M., Clarelli, F., ... & De Bellis, G. (2016). Next generation sequencing of pooled samples: guideline for variants’ filtering. Scientific reports, 6, 33735.

3) Estimates of genetic differentiation Fst: The manuscript main objective is to validate the use of pool-seq by comparing it with mtCR of a published study (Clarke et al. 2015), yet in the main Figures of the manuscript there is not a single Figure that clearly shows and highlight the results of this comparison. The results are distributed between Figure 2 and table S2. I think its important to combine these two into Figure 2 so the reader has an immediate access to this comparison which is the main core of the study.


- Minor and more specific comments:

• Abstract:
Line-38: In this line where you mention “thousands” of nuclear markers, I suggest to replace “thousands” by the exact number of nuclear markers that you have actually used after filtration.
• Introduction:
Line-49: Many elasmobranchs through out the ocean; the reference of the ocean here is quite vague and does not add value, I suggest you either drop the ocean or be more specific, for example; Many elasmobranchs through out the five ocean basins…
Line-72: the reference Domingues et al. 2017a: I believe the “a” is left by mistake as I only see one Domingues et al. 2017 in the reference list.
Line-106: I understand that within the scope of this paper you are focused on evaluating pool-seq functionality compared to mtCR, as these were the data provided by Clarke et al. 2015 and you are using the same individuals. But as you mentioned earlier in your introduction, the combination of mtDNA and microsatellites for the assessment of population structure in migratory species is essential. Could you add a sentence here acknowledging that; in this study you are only focused on mtDNA because it was the tool tested by Clarke et al. 2015. Also, it could be useful to add a sentence mentioning that one of the additional advantages of using pool-seq is the fact that through one tool you would be able to analysis both mtCR and nuclear markers, SNPs in this case, which are also easier to score and analyze than microsatellite (e.g more prone to genotyping errors effecting the analysis through; null allele, slippage, or allele drop-out). I think this could be another reason to justify and support your call to use pool-seq as an alternative tool.
Line-123: Again, Domingues et al. 2017b, there is only one Domingues et al. 2017 in the reference list!!
Line-124: At the end of this paragraph, could you provide a sentence explaining why you think Domingues et al. 2017 showed different results than Clarke et al. 2015 between the north and south Western Atlantic. (i.e. possibly due to sampling higher number of samples and the south western Atlantic was represented by a further south samples).
• Methods:
Line-152: the sentence “Number of individuals per samples are displayed in Fig 1”, what do you mean by “per samples”? Sample pool? Also the number of samples obtained from Clarke et al. 2015 is the same as the listed samples in the map, however you have dropped some samples that showed degradation or were overly digested from the pool-seq. Could you add a sentence to clarify the net number of the used samples per location/pool after dropping the degraded samples.
Since pool-seq is based on estimates of the allele frequencies of the pooled samples, and you do not obtain individual genotyping data that are needed to run the basic assessment of neutrality (HWE and LD), I would suggest to generate a figure showing the overall distribution of allele frequency from each pool and evaluate their distribution trends compare to each other. For your SNP outlier test, I assume it was a PCA assessment? If what you have performed is a PCA outlier test, I think having a figure in the supplementary on the distribution of allele frequency from each pool is useful and will give the reader more information to evaluate the state at each pool. I believe all your selected SNPs are bi-allelic so the expected HWE distribution for your SNP markers would be a binomial distribution, would be interesting to see how they deviate from that and if this deviation is the product of structure or genotyping error (compare the distribution between structured groups compared to overall).
• Results:
Line-236: 30 SNPs is quite a low number, another reason why changing MAF to 1% could add more information to the analysis.
Figures and tables:
- Figure-2: As mentioned earlier, I think it would be very useful to modify this figure to incorporate Fst values generated by Clarke et al 2015. I am aware that you have mentioned it in the manuscript text, but having it in a figure is also useful.
- Fig.2: RD or RS?!!
Supplementary data:
- I have noticed that your supplementary tables do not have headers and figures were provided with out captions, could you kindly add this information.
• Discussions:
- Line-286: rephrase: “Pool-seq was able to detect isolated populations within the Gulf of Mexico, in the Western Atlantic, and along the Brazilian coast, where Clarke et al. (2015) found no population structure”. Please correct me if I am wrong, my understanding is that you had one sample site from Gulf of Mexico? While in this statement you mention that you detect isolated populations “within the Gulf of Mexico” is that righ?

I think it would be important to clearly mention in few sentences at the end of your discussion that pool-seq is a cost effective tool but is limited. For high budget projects its more reliable and informative to other tools such as RADseq and SNP array which are individual based genotyping methods that will allow deeper assessment of the sampled populations. However, pool-seq is suited for cases that needs rapid and cost effective assessment to serve the immediate and extensive need in providing population structure studies to serve their management.
• Acknowledgment:
Please provide what is supposed to be filled in place of “#XXXX” in line 380 and 381.

---

## Round 0.2 · accepted · Accept

· Academic Editor

Accept

Please correct according to the Reviewer's remark on "XXXXX" (Line 435).

·

Basic reporting

no comment

Experimental design

no comment

Validity of the findings

no comment

Additional comments

The manuscript really improved during the review proccess and as far as I'm concerned it is good to go.
Do not forget to change the "XXXXX" (Line 435) for the correct number.
Warm regards
Cesar

·

Basic reporting

Kraft et al. present a manuscript entitled “Genomics versus mtDNA for resolving stock structure in the silky shark (Carcharhinus falciformis)”, where they tested the use of pool-seq as an alternative, cost effective and more sensitive tool for the delination of the C. falciformis stock structure.

Experimental design

The manuscript has clearly stated the research question, the gap intended to fill which fits within the scope of PeerJ. Methods are well explained and reproducible.

Validity of the findings

Conclusions are clearly stated and connected to the main question investigated.

Additional comments

Thank you for providing a clear response, your work will provide a valuable addition towards the evaluation of genetic structure in elasmobranchs. All the best!